# Gene editing and elimination of latent herpes simplex virus in vivo

Martine Aubert [1], Daniel E. Strongin[2], Pavitra Roychoudhury [2], Michelle A. Loprieno[1], Anoria K. Haick[1], Lindsay M. Klouser[1], Laurence Stensland[2], Meei-Li Huang[2], Negar Makhsous[2], Alexander Tait[2], Harshana S. De Silva Feelixge[1], Roman Galetto[3], Philippe Duchateau [3], Alexander L. Greninger[2], Daniel Stone [1] & Keith R. Jerome [1,2✉]

We evaluate gene editing of HSV in a well-established mouse model, using adeno-associated virus (AAV)-delivered meganucleases, as a potentially curative approach to treat latent HSV infection. Here we show that AAV-delivered meganucleases, but not CRISPR/Cas9, mediate highly efficient gene editing of HSV, eliminating over 90% of latent virus from superior cervical ganglia. Single-cell RNA sequencing demonstrates that both HSV and individual AAV serotypes are non-randomly distributed among neuronal subsets in ganglia, implying that improved delivery to all neuronal subsets may lead to even more complete elimination of HSV. As predicted, delivery of meganucleases using a triple AAV serotype combination results in the greatest decrease in ganglionic HSV loads. The levels of HSV elimination observed in these studies, if translated to humans, would likely significantly reduce HSV reactivation, shedding, and lesions. Further optimization of meganuclease delivery and activity is likely possible, and may offer a pathway to a cure for HSV infection.

---

[1] Vaccine and Infectious Disease Division, Fred Hutchinson Cancer Research Center, Seattle, WA, USA. [2] Department of Laboratory Medicine, University of Washington, Seattle, WA, USA. [3] Cellectis SA, Paris, France. ✉email: kjerome@fredhutch.org

Herpes simplex virus (HSV) types 1 and 2 are widespread and important human pathogens, causing oral and genital ulcers, neonatal herpes, and increasing the risk of acquiring HIV. After primary infection at the skin or mucosa, HSV establishes lifelong latency in both sensory (e.g., trigeminal and dorsal root ganglia) and autonomic (e.g., superior cervical and major pelvic ganglia) neurons of the peripheral nervous system. HSV can subsequently reactivate from the latent state, causing lesions and/or virus shedding at mucosal surfaces[1–7]. While current antiviral therapies reduce the severity of acute infections and diminish viral reactivation frequency[8–12], they do not reduce or eliminate the latent virus that drives recurrent disease. Gene editing using CRISPR/Cas9, meganucleases, or similar enzymes offers the possibility of directly targeting latent genomes for disruption or elimination while preserving neurons, thus eliminating the possibility of viral reactivation and pathogenesis[13–20].

In vivo gene editing of latent HSV genomes within TG sensory neurons of mice has been previously demonstrated using HSV-specific meganucleases delivered via adeno-associated virus (AAV) vectors, but the levels of gene editing were modest (<4%)[13]. In an effort to improve endonuclease-directed gene editing of latent HSV genomes to levels needed for therapeutic benefit, we have evaluated the use of improved self-complementary (sc)AAV vectors, simultaneous targeting of multiple sites within the HSV genome, substitution of CRISPR/Cas9 for meganucleases, and the relative distribution of HSV vs. AAV vectors at the single-cell level. Our results provide critical insights for the optimization of in vivo gene therapy against HSV, and suggest that meganuclease-mediated gene editing represents a plausible pathway toward HSV cure.

## Results

**Gene editing reduces ganglionic HSV.** To evaluate the impact of efficient meganuclease-mediated gene editing on latent HSV infection in vivo, we used a mouse model of HSV ocular infection as previously described[13] (Fig. 1a). We used the HSV-specific meganucleases HSV1m5 (m5), targeting $U_L19$ which codes for the major capsid protein VP5; HSV1m8 (m8), targeting $U_L30$ which codes for the catalytic subunit of the viral DNA polymerase, and HSV1m4 (m4), targeting the duplicated gene ICP0[13,17,20]. Initially, latently infected mice were treated using an scAAV8 delivery vector, either as single-meganuclease (m5 or m8) or dual-meganuclease (m5 + m8) therapy, at the indicated dose (Fig.1a). A month later, mice were sacrificed, and superior cervical (SCG) and trigeminal (TG) ganglia collected for analysis. In agreement with our previous results[13], animals receiving a single meganuclease (either m5 or m8) showed modest levels of gene editing of HSV target sites (means of 4.7% for m5 and 0.97% for m8 in SCG, and 0.92% for m5 and 0.17% for m8 in TG), and neither SCG nor TG showed a detectable reduction in ganglionic HSV load compared with control mice (Fig. 1b). However, when infected mice were treated with dual-meganuclease (m5 + m8) therapy, a significant decrease in HSV loads was detected in both SCG and TG, with a mean level of HSV genomes/$10^6$ ganglionic cells in SCG of $7.3 \times 10^3$ in dual-meganuclease-treated mice compared with $3.5 \times 10^4$ in control animals (79% reduction, $p = 0.018$), and in TG of $9.6 \times 10^3$ in dual-meganuclease-treated mice compared with $4.2 \times 10^4$ in control animals (77% reduction, $p = 0.001$) (Fig. 1b). Interestingly, the HSV genomes remaining after dual-meganuclease therapy had mean gene editing levels similar to that of single-nuclease treated mice (3.6% for m5 and 0.63% for m8 in SCG, and 1.0% for m5 and 0.21% for m8 in TG of dual-meganuclease-treated mice, Fig. 1c, d). In both single- and dual-meganuclease-treated mice, the pattern of mutation (mostly small

deletions of 1–16 bp) seen was consistent with that previously observed in gene editing of HSV[13,20].

To evaluate the role of AAV serotype in gene editing efficacy, latently infected mice were treated with single-nuclease therapy delivered by a vector derived from the AAV-Rh10 serotype (Fig. 2a). This serotype had demonstrated efficient ganglionic transgene delivery in optimization studies, which showed that the AAV-Rh10 serotype administered via retro-orbital injection led to the highest levels of HSV gene editing in ganglia (Supplementary Fig. 1). While reduction of viral load in TG did not reach statistical significance as previously seen with dual-meganuclease therapy (Fig. 1b), a ~65% reduction in mean HSV load in SCG was detected (mean $6 \times 10^3$ HSV genomes/$10^6$ ganglionic cells in treated mice, compared with a mean of $3.7 \times 10^4$ HSV genomes/$10^6$ ganglionic cells in control animals, $p = 0.017$) (Fig. 2b). Interestingly, the HSV genomes remaining after single-meganuclease therapy delivered by scAAV-Rh10 (Fig. 2c) showed up to 30% mutagenesis by NGS, a level substantially higher than observed after dual-meganuclease therapy.

We then evaluated dual-meganuclease therapy delivered by AAV serotype Rh10, consisting of $5 \times 10^{11}$ vector genomes (vgs) of scAAV-Rh10-CBh-m5 and $5 \times 10^{11}$ vgs of scAAV-Rh10-CBh-m8 (Fig. 2d). Dual-meganuclease therapy (m5 + m8) delivered by scAAV-Rh10 resulted in a significant decrease of latent HSV genomes in both SCG (mean $2 \times 10^4$ vs. $2.6 \times 10^3$, 86% reduction, $p = 0.0006$) and TG (mean $3.9 \times 10^4$ vs. $2.1 \times 10^4$, 45% reduction, $p = 0.01$) of treated mice compared with untreated controls (Fig. 2e). Among the virus remaining after therapy, the mean levels of target site mutation were 4.2% for m5 and 0.4% for m8 in SCG, and 3.8% for m5 and 0.4% for m8 in TG (Fig. 2f).

Taken together, these results suggest that while single DNA double strand breaks (DSB) in HSV are typically repaired, often resulting in mutation, the creation of two DNA DSB more commonly leads to degradation and loss of HSV genomes. To evaluate further the importance of one compared to two or more DSB, we injected HSV latently-infected mice with $10^{12}$ vgs of scAAV-Rh10 expressing HSV1m4 (m4), a meganuclease targeting a sequence in the duplicated gene coding for ICP0, which therefore induces two DNA DSB in HSV (Fig. 3a)[17]. A significant decrease in HSV genomes was detected in both SCG (59% decrease, $p = 0.03$) and TG (45% decrease, $p = 0.02$) of m4-treated mice compared with untreated control animals (Fig. 3b). Consistent with the results from dual m5 + m8 meganuclease-treated animals, by NGS analysis only 1.6% (SCG) and 0.8% (TG) of target sites in remaining viral DNA were mutated in m4-treated mice (Fig. 3c), again supporting the interpretation that creation of two DNA DSB preferentially leads to degradation of HSV DNA.

To evaluate whether the introduction of more than two DNA DSB would further improve the elimination of HSV genomes by gene editing (Fig. 3d), latently infected mice were administered AAV-Rh10/dual-meganuclease therapy, consisting of $5 \times 10^{11}$ vector genomes (vgs) of scAAV-Rh10-CBh-m5 (which targets a single site in HSV) and $5 \times 10^{11}$ vgs of scAAV-Rh10-CBh-m4 (which targets two additional sites). While a reduction in HSV viral load was observed in treated animals compared with controls (82.5% in SCG, $p = 0.00001$; 22.5% in TG, $p = 0.47$) (Fig. 3e), along with mutations in the remaining genomes at both the m5 (on average 5.7% in SCG; 2.7% TG) and m4 (on average 1.9% in SCG; 1.7% TG) sites (Fig. 3f), these did not appear to be superior to dual m5 + m8 or m4 therapy. The target site mutation frequencies observed were higher for m5 than m4 or m8, which is consistent with a previous report that this enzyme has a higher activity[17].

To determine the impact of dual-meganuclease treatment on the ability of HSV to reactivate from ganglia of treated mice, we

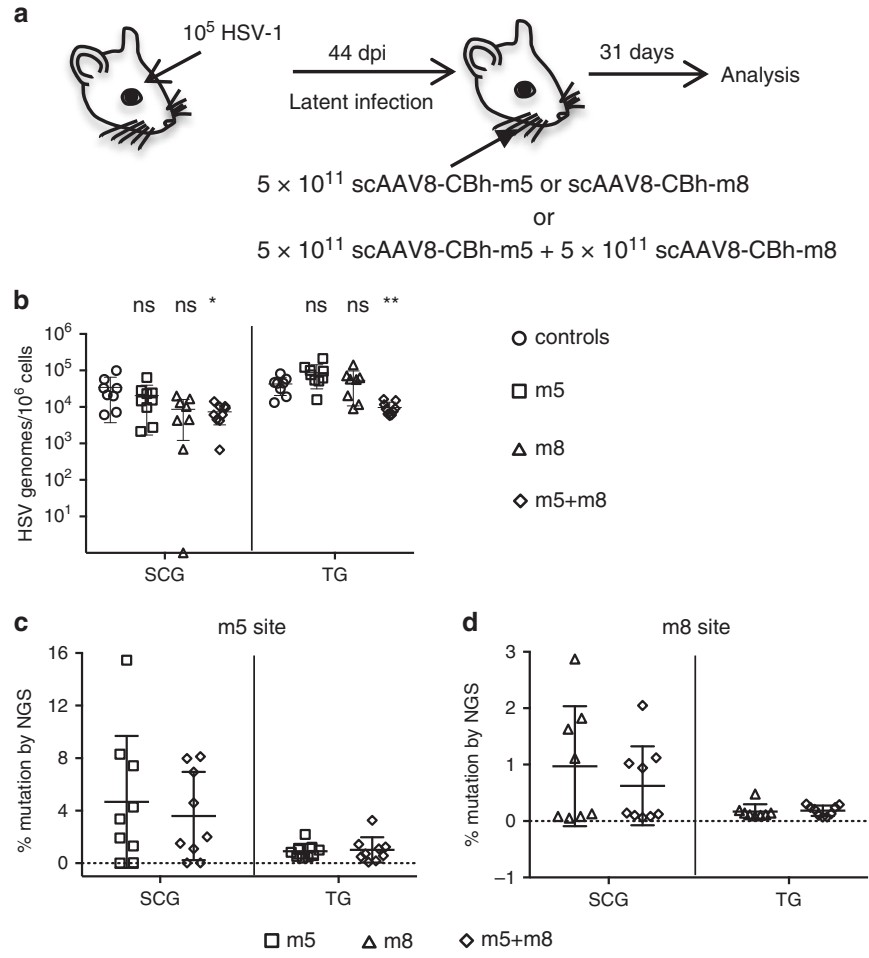

**Fig. 1 HSV load reduction in dual-meganuclease-treated mice. a** Mice latently infected for 44 days with $10^5$ PFU HSV-1 17+ were left untreated (Controls, circles $n = 8$) or administered by whisker pad injection either $5 \times 10^{11}$ vgs scAAV8-CBh-m5 (squares, $n = 9$), $5 \times 10^{11}$ vgs scAAV8-CBh-m8 (triangles, $n = 8$) or $5 \times 10^{11}$ vgs of each scAAV8-CBh-m5 and scAAV8-CBh-m8 (diamonds, $n = 9$). Analysis was performed 31 days later. **b** ddPCR quantification of HSV genomes in SCG and right (ipsilateral) TGs from infected mice ($p = 0.018$ in SCG of controls vs m5 + m8; $p = 0.001$ in TG of controls vs m5 + m8). **c**, **d** NGS analysis of m5 and m8 sites in HSV genomes from SCGs and right (ipsilateral) TGs from infected mice treated with either m5 (squares, $n = 9$), m8 (triangles, $n = 8$) or m5 + m8 (diamonds, $n = 9$). ns: not significantly different from controls, *$p < 0.05$, **$p < 0.01$, ***$p < 0.001$, significantly different from controls. All data are presented as mean values ± SD. Statistical analysis was conducted using unpaired multiple $t$-test without correction for multiple comparison.

collected TG and SCG from dual-AAVRh10/meganuclease-(m5 + m8) treated and control mice, and subjected the tissues to explant reactivation for 24 h (Fig. 4a) as previously described[21]. We have shown that ganglionic explant reactivation resulted in an approximate two to three-fold increase in total HSV levels over fresh ganglia, which can be measured by ddPCR (Supplementary Fig. 2a, b). Consistent with our results in Fig. 2, a significant decrease in HSV genomes in reactivated ganglia was detected in dual-meganuclease-treated mice compared with untreated animals in both SCG (90% reduction, mean $4.9 \times 10^4$ vs $4.2 \times 10^3$, $p = 0.002$), and TG (51% reduction, mean $8.9 \times 10^4$ vs $4.4 \times 10^4$, $p = 0.01$) (Fig. 4b). By NGS, gene editing was detected in 6.0% of residual virus for HSV1m5 and 1.4% for HSV1m8 in SCG, and 3.9% for HSV1m5 and 0.3% for HSV1m8 in TG (Fig. 4c). Strikingly, these decreases in total HSV in reactivated ganglia resulted in a 95% (SCG) and 55% (TG) reduction of de novo produced HSV genomes from dual-treated ganglia compared with untreated control (Fig. 4d–e and Table 1).

**AAV-Cas9 mediates only weak gene editing of HSV in vivo.** To investigate whether the CRISPR/Cas9 system might allow more

efficient gene editing of HSV than meganucleases, we identified several *Staphylococcus aureus* (Sa)Cas9 sgRNAs targeting two essential HSV genes: $U_L54$ encoding the immediate-early regulatory protein ICP27 (sgRNA$_{UL54}$ 13, 17, and 26) and $U_L30$ (also the target for meganuclease HSV1m8) coding for the catalytic subunit of the viral DNA polymerase (sgRNA$_{UL30}$ 1 and 10). Due to the large SaCas9 coding sequence (3.1 kb), ssAAV was used for the delivery system. The larger payload capacity of ssAAV allowed both SaCas9 and sgRNA expression cassettes to be on the same AAV construct, ensuring simultaneous delivery of SaCas9 and sgRNA to transduced cells. Several sgRNAs were able to promote high-level Cas9 gene editing of HSV genomes in latently-infected cultured neurons transduced with SaCas9/ sgRNA-expressing AAV vectors, as detected by T7 endonuclease 1 (T7E1) assay (Fig. 5a–c). By more-sensitive next-generation sequencing (NGS) analysis, the highest level of mutation detected was 49% when SaCas9 was paired with sgRNA$_{UL54-26}$ (Supplementary Table 1).

To evaluate the ability of Cas9 to gene edit HSV in vivo, we established latent HSV infection in mice by ocular infection as above[13]. Thirty days after HSV infection, mice were administered $10^{12}$ vgs of ssAAV1-sCMV-SaCas9-sgRNA$_{UL54}$ via whisker pad

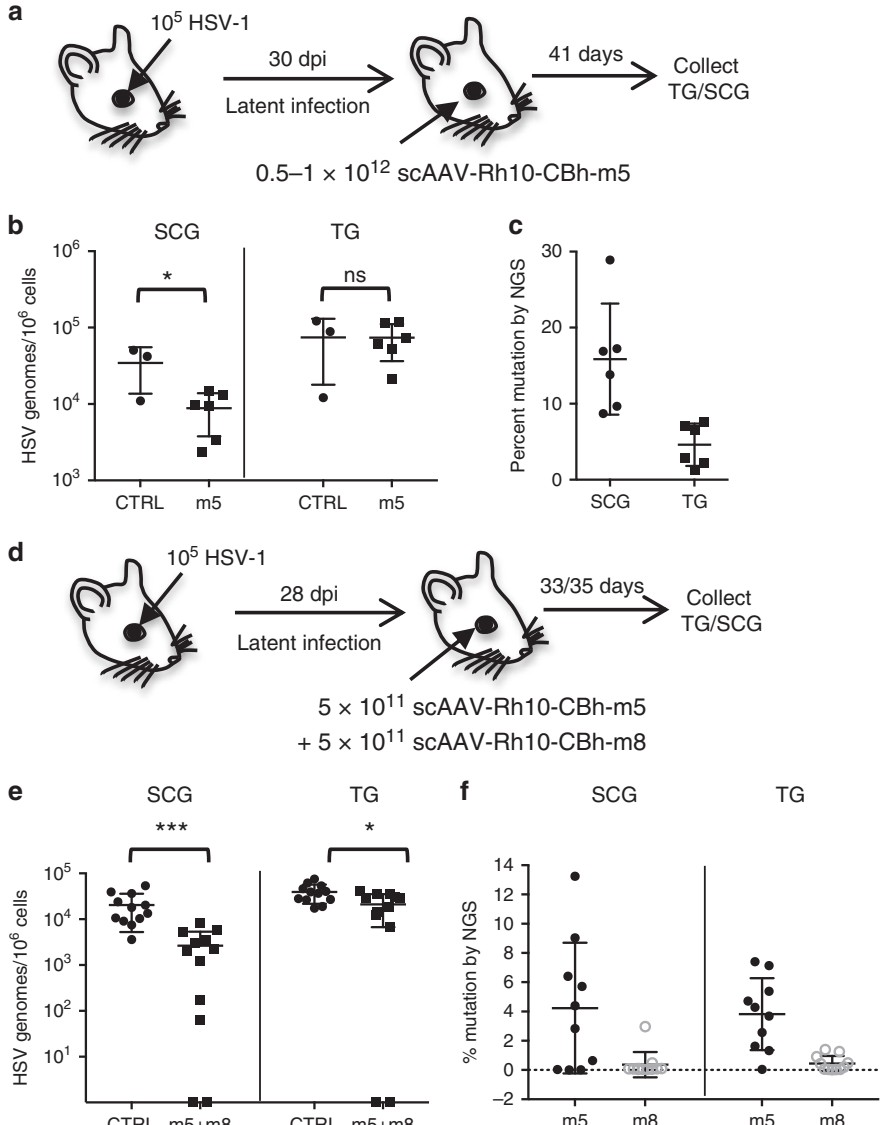

**Fig. 2 Efficient gene editing after optimized delivery of dual-meganuclease therapy. a** Mice latently infected with $10^5$ PFU HSV-1 17+ for 30 days, were administered 0.5–1 × $10^{12}$ scAAV-Rh10-CBh-m5 by RO injection. Analysis was performed 41 days later. **b** ddPCR quantification of HSV genomes in SCG and right (ipsilateral) TG of control (CTRL, circles $n = 3$) or m5 (squares $n = 6$) treated mice. $p = 0.017$ for SCG. **c** NGS analysis in SCG and TG from m5-treated mice ($n = 6$) to detect HSV gene editing at the m5 target site in SCG (circles) or TG (squares). **d** Mice latently infected with $10^5$ PFU HSV-1 17+ in the right eye for 28 days were administered 5 × $10^{11}$ vgs scAAV-Rh10-CBh-m5 + 5 × $10^{11}$ vgs scAAV-Rh10-CBh-m8 by RO injection. SCGs and ipsilateral TG were collected 33–35 days later. **e** ddPCR quantification of HSV genomes in TG and SCG from latently infected mice either left untreated (Controls, CTRL circles, $n = 12$) or administered m5 + m8 (squares, $n = 12$). $p = 0.006$ and $p = 0.01$ for SCG and TG, respectively. **f** NGS analysis of SCG and TG from treated mice to detect mutations at either the m5 (closed circles) or m8 (open circles) target sites. ns: not significantly different from controls, *$p < 0.05$, **$p < 0.01$, ***$p < 0.001$, significantly different from controls. All data are presented as mean values ± SD. Statistical analysis was conducted using unpaired multiple $t$-test without correction for multiple comparison. Source data are provided as a Source data file.

injection, and TG were collected at 28 and 56 days post-injection for analysis (Fig. 5d). In agreement with our results with single-meganuclease therapy, quantification by ddPCR showed similar levels of HSV in the TG of treated and control animals (Fig. 5e). However, in contrast to the easily detected gene editing of HSV after single-meganuclease therapy, we were unable to detect gene editing of HSV in any of the treated mice by T7E1 assay (Supplementary Fig. 3a–c), despite AAV loads equal to or higher than those observed in our previous experiments (Supplementary Fig. 4a). By more-sensitive NGS analysis, we were able to detect only very low levels of mutation (0.1–0.3%) in the Cas9-treated animals (Fig. 5f).

Although the sCMV promoter used in the above experiments with Cas9 can mediate strong transgene expression in sensory neurons in vitro[20] and in vivo[22], we considered the possibility that optimization of the promoter might allow more efficient gene editing of latent HSV by Cas9. We, therefore, performed similar in vivo experiments (Supplementary Fig. 5a) using ssAAV1 vectors expressing SaCas9 under control of the alternative strong constitutive promoters CMV, nEF, and CBh. Along with the most effective sgRNA targeting $U_L54$ (sgRNA$_{UL54-26}$), we tested two additional sgRNAs targeting $U_L30$ (sgRNA$_{UL30-1}$ and sgRNA$_{UL30-10}$). We also assessed the alternative AAV8 serotype, which achieved easily detectable gene editing of HSV when delivering meganucleases

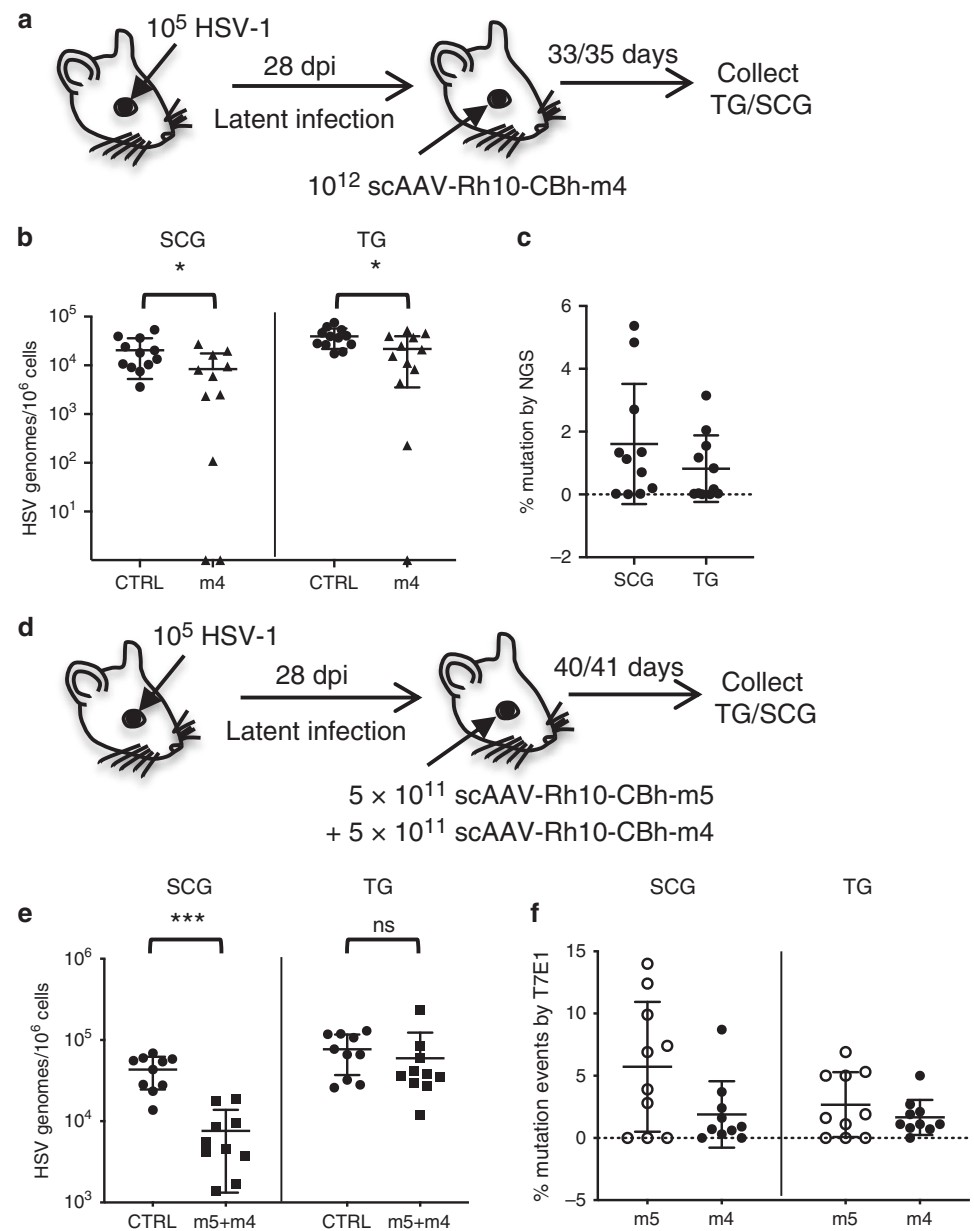

**Fig. 3 Reduction of ganglionic HSV genomes after induction of multiple DSB. a** Mice latently infected with $10^5$ PFU HSV-1 17+ for 28 days, were either left untreated (Controls, CTRL, n = 12) or administered $1 \times 10^{12}$ vgs scAAV-Rh10-CBh-m4 (m4, n = 12) by RO injection. These mice were infected and treated at the same time as the mice described in Fig. 2d–f (control mice are the same for these 2 data sets). At 33–35 days post meganuclease-therapy, SCGs and right (ipsilateral) TG were harvested and **b** HSV genomes quantified by ddPCR. p = 0.03 and p = 0.02 for SCG and TG, respectively. **c** NGS analysis of SCG and TG from dual-meganuclease-treated mice to detect HSV gene editing at the m4 target site in SCG or TG. **d** Mice latently infected with $10^5$ PFU HSV-1 17+ for 28 days were either left untreated (Controls, CTRL, n = 10) or administered $5 \times 10^{11}$ vgs scAAV-Rh10-CBh-m5 + $5 \times 10^{11}$ vgs scAAV-Rh10-CBh-m4 (m5 + m4, n = 10) by RO injection. At 40–41 days post meganuclease-therapy, SCGs and right (ipsilateral) TG were harvested from infected mice either untreated (closed circles) or treated with m5 + m4 (closed squares) and **e** HSV genomes quantified by ddPCR, p = 0.00001 for SCG. **f** NGS analysis of SCG and TG from dual-meganuclease-treated mice to detect HSV gene editing at m5 (open circles) and m4 (closed circles) target sites in SCG or TG. ns: not significantly different from controls, *p < 0.05, **p < 0.01, ***p < 0.001, significantly different from controls. All data are presented as mean values ± SD. Statistical analysis was conducted using unpaired multiple t-tests without correction for multiple comparison. Source data are provided as a Source data file.

(Fig. 1a–d). Quantification by ddPCR showed similar levels of HSV in the TG of all treated animals (Supplementary Fig. 5b, c). As observed in the previous experiments, no gene editing of latent HSV genomes was detected by the T7E1 assay under any conditions using Cas9 (Supplementary Fig. 5f–h). In agreement with this and our previous results, NGS analysis demonstrated that levels of gene editing were very low, and were observed in only a subset of animals (Supplementary Fig. 5i, j).

Similar to our results with meganucleases, CRISPR/Cas9 has shown significantly higher gene disruption efficiency when targeting dual sites[23–25]. We therefore tested whether dual sgRNA therapy could result in higher gene editing of latent HSV. Latently infected mice were administered either dual sgRNA therapy consisting of $10^{12}$ vg of ssAAVRh10-sCMV-Cas9-sgRNA$_{UL54-26}$ and $10^{12}$ vg of ssAAVRh10-sCMV-Cas9-sgRNA$_{UL30-10}$ or single-meganuclease therapy with $10^{12}$ vg of

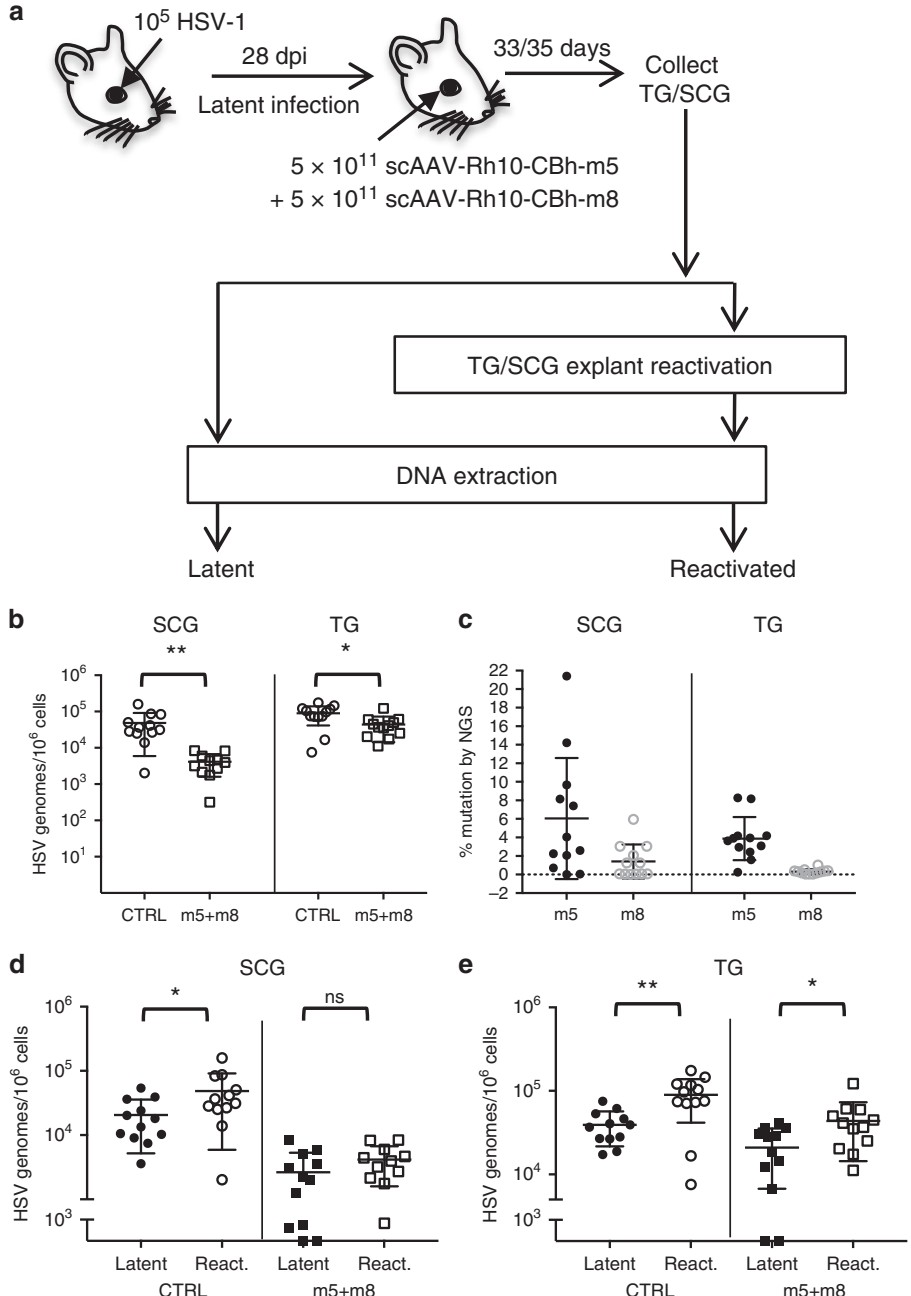

**Fig. 4 Reactivation after dual-meganuclease therapy. a** Ganglia from a second set of mice latently infected and treated with dual-meganuclease therapy at the same time as those described in Fig. 2d–f, were subjected to ganglia (SCG/TG) explant reactivation (see "Methods") prior to DNA extraction. **b** ddPCR quantification of HSV genomes in reactivated SCG and TG from latently infected untreated control mice (CTRL, open circles, $n = 12$) or dual-meganuclease-treated mice (open squares, $n = 12$). $p = 0.002$ and $p = 0.01$ for SCG and TG, respectively. **c** NGS analysis in reactivated SCG and TG from dual-meganuclease-treated mice to detect HSV gene editing at either the m5 (m5, closed circles) or m8 (m8, open circles) target sites. **d**, **e** Comparison of HSV loads in SCG and TG from latently infected (closed circles) and reactivated (open circles) control mice (CTRL), $p = 0.02$ for SCG, and from latently infected (closed squares) or reactivated (open squares) dual-meganuclease-treated mice (m5 + m8), $p = 0.0012$ and $p = 0.012$ for **d** SCG and **e** TG, respectively. *$p < 0.05$; **$p < 0.01$; ns: not significantly different. All data are presented as mean values ± SD. Statistical analysis was conducted using unpaired multiple $t$-tests without correction for multiple comparison (**c**) and one-tail, unpaired $t$-test (**d**, **e**). Source data are provided as a Source data file.

ssAAVRh10-smCBA-HSV1m5-Trex2-mCherry (Fig. 6a). The ssAAV construct carrying the HSV1m5 also delivers Trex2, a 3′–5′ exonuclease that was shown previously to increase meganuclease gene editing[20]. No loss of HSV genomes was observed in the ganglia of either dual sgRNA/cas9- or single-meganuclease-treated mice compared to control animals (Fig. 6b, c). NGS analysis showed that gene editing was seen in some but not all treated animals regardless of the therapy received, and the levels of mutation observed in ganglia from dual sgRNA treated mice remained weak and lower (<0.2%) than those from ganglia of single-meganuclease-treated mice (up to 9.9% in SCG and 1.1% in TG; Fig. 6d, e). RNA expression of Cas9, sgRNA, and m5 was tested to determine whether low enzyme expression could explain the weak gene editing. While Cas9 mRNA was detected in 80% of the SCG and TG from dual sgRNA/cas9-treated mice, only 40 and 20% of mice had detectable levels of sgRNA in SCG

**Table 1 HSV loads in latent and reactivated ganglia after dual-meganuclease therapy.**

| Tissue | Latent HSV in CTRL[a] (×10³) | HSV after reactivation in CTRL[a] (×10³) | Fold increase in HSV in CTRL[b] | De novo produced HSV in CTRL (×10³) | Latent HSV in m5 + m8 treated[d] (×10³) | HSV after reactivation in m5 + m8 treated[d] (×10³) | Fold increase in HSV in m5 + m8 treated[b] | De novo produced HSV in m5 + m8 treated (×10³) | % Reduction in de novo produce HSV[c] |
|---|---|---|---|---|---|---|---|---|---|
| SCG | 20.5 | 48.8 | 2.4 | 28.3 | 2.64[a] | 4.16[a] | 1.6 | 1.52 | 94.6 |
| TG | 39.3[d] | 89.9[d] | 2.3 | 50.6 | 21.0[d] | 43.8[d] | 2.1 | 22.8 | 54.9 |

All data are presented as mean values, and statistical analysis was conducted using one-tail, unpaired t-test. Source data are provided as a Source Data file.
[a]Mean HSV loads in SCG or TG from untreated control (CTRL) mice with latent (n =12) or reactivated (n =12) virus.
[b]Fold increase in HSV genomes after reactivation in SCG or TG from either untreated control (CTRL) mice or dual-meganuclease (m5 + m8) treated mice.
[c]Percent reduction in de novo produced HSV genomes in dual-meganuclease-treated compared with untreated control tissues.
[d]Mean HSV loads in SCG or TG from dual-meganuclease (m5 + m8) treated mice with latent (n =12) or reactivated (n =12) virus.

and TG, respectively (Fig. 6f–i). For comparison, with single-meganuclease therapy, expression of HSV1m5 was detected in only 50% of the TG and SCG of treated animals, despite the easily detectable gene editing (Fig. 6j, k).

**Single-neuron analysis of HSV and AAV vectors.** While optimizing AAV-meganuclease therapy, we obtained different levels of HSV gene editing in ganglia depending on the AAV serotype used for the meganuclease delivery. The route of administration of the AAV delivery vectors influenced the efficiency of HSV gene editing. Delivery of AAV to the sites of HSV latency was shown previously to be less efficient after administration via the cornea (with or without scarification) than intradermal whisker pad (WP) injection[22]. In our optimization studies presented in Supplementary Fig. 1, we showed that while AAV delivery to ganglia was similar after injection via RO, TV, and WP injection, RO led to the highest levels of gene editing, especially with Rh10 serotype. Furthermore, gene editing and loss of viral genomes were consistently greater in SCG than TG (Figs. 1–4, Supplementary Fig. 1). We hypothesized that these differences might result from the dissimilar distribution of AAV vector serotypes and HSV between TG and SCG, and among the different types of neurons within ganglia. To evaluate this issue, we assessed AAV-mediated gene delivery to HSV-infected neurons in mice using single-cell RNA-sequencing (scRNA-seq). Mice were latently infected with HSV-1, after which each latently infected mouse received $10^{12}$ vgs of one of four AAV vector serotypes reported to possess neuronal tropism in mice[22,26,27]. Each AAV serotype carried a unique marker transgene: AAV1-mScarlet; AAV8-mEGFP; AAV-PHP.S-DsRed-Express2; and AAV-Rh10-mTagBFP-2. Three weeks after AAV injection, TG and SCG were collected and TG or SCG pooled from all animals for neuron purification, library construction, and sequencing. We obtained high quality single-cell expression data from 2319 purified TG neurons and 2041 SCG neurons (99,817 mean reads and 5,908 median genes per cell for TG; 94,797 mean reads and 5635 median genes per cell for SCG).

To determine whether specific subtypes of neurons were infected by HSV and targeted by each AAV serotype, we first classified neurons into groups based on gene expression profiles. Principal component analysis (PCA) of the single-cell gene expression data identified transcriptionally distinct clusters of neurons for each tissue: 5 for the SCG and 10 for the TG. As expected, SCG (autonomic) and TG (sensory) neurons fully segregated from one another (Fig. 7a and Supplementary Fig. 6a). The cell clusters in the TG correlated well with clusters identified previously in the DRG[28,29] and the TG[30]. For example, cluster TG-8 corresponds closely to Cluster 6 in Nguyen et al.[30], cluster PEP2 in Usoskin et al.[28], and cluster C8-2 in Li et al.[29] (Supplementary Fig. 6b–e).

In order to determine the relative distribution of HSV and each AAV serotype across neuronal subtypes, we identified neurons that expressed HSV transcripts, as well as each of the marker transgenes carried by the four different AAV serotypes (mScarlet (AAV1), mEGFP (AAV8), DsRed-Express2 (AAV-PHP.S), and mTagBFP-2 (AAV-Rh10)) (Supplementary Fig. 7, Supplementary Tables 2–3). The latency-associated transcript (LAT), the only HSV RNA highly expressed during latent infection (reviewed in ref. [1]), accounted for >99% of all HSV transcripts detected. HSV transcripts were detected in 1.4% of cells in the SCG and 12.2% of cells in the TG, consistent with our ddPCR results that showed 10-fold fewer HSV genomes in the SCG (Supplementary Fig. 1). HSV-expressing cells were non-randomly distributed across the different neuronal clusters within both the SCG ($\chi^2$, $p < 0.0001$) and the TG ($\chi^2$, $p < 0.0001$). For example, in the SCG,

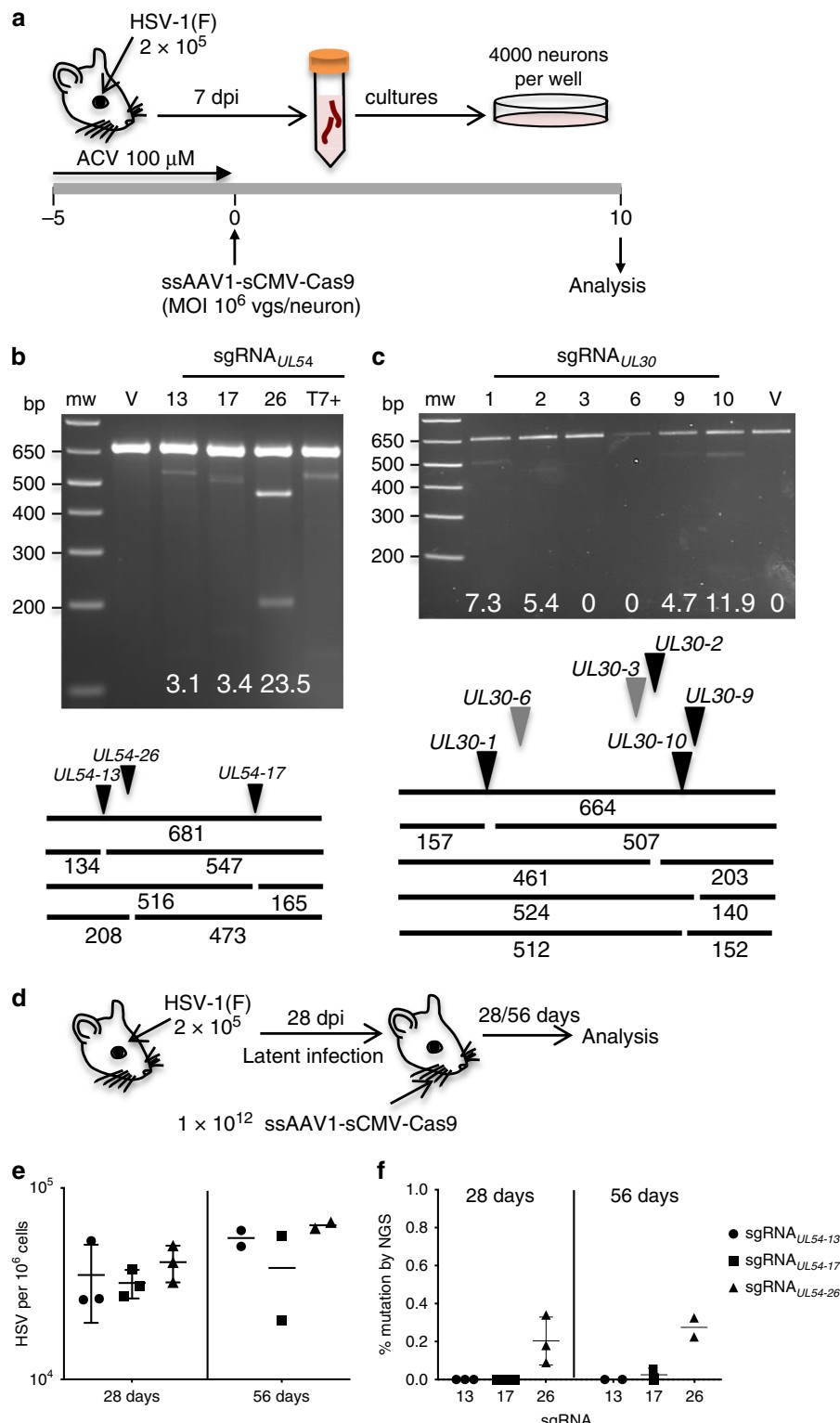

HSV-expressing cells were most enriched in SCG-4, and absent from SCG-3, whereas in the TG HSV-expressing cells were most enriched in TG-8 (Fig. 7b, c, Supplementary Figs. 8–9, Supplementary Tables 2–3).

The distribution of transgene expression from AAV varied by serotype and tissue. AAV1 (mScarlet) supported minimal expression in the SCG, with only 2.3% of neurons expressing mScarlet, but supported the broadest transgene expression in the TG (18.2% of cells). By contrast, mEGFP (AAV8) was expressed in 60.4% of SCG neurons and 11.9% of TG neurons, DsRed-Express2 (PHP.S) was expressed in 18.9% and 12.3% of neurons in the SCG and TG respectively, and mTagBFP-2 (Rh10) was expressed in 23.5% and 9.6% of SCG and TG neurons respectively (Supplementary Fig. 7). Within the SCG, mScarlet (AAV1)-expressing cells were randomly distributed across the clusters, while mEGFP (AAV8), DsRed-Express2 (AAV-PHP.S) and mTagBFP-2 (AAV-Rh10) exhibited a non-random distribution ($\chi^2$: $p < 0.0981$, $p < 0.0001$, $p = 0.0143$, and $p < 0.0001$, respectively;

**Fig. 5 SaCas9 gene editing of HSV in infected neuronal cultures. a** Schematic of neuronal culture generation and exposure to AAV/CRISPR-Cas9 treatment. Mice were infected with $2 \times 10^5$ PFU of HSV-1(F); right TGs were collected 7 days later. Neuronal cultures were established, and cells were cultured for 5 days in medium supplemented with 100 μM ACV as previously described[13]. Cells were then transduced at the indicated time at a MOI of $10^6$ AAV vector genomes per neuron, with either ssAAV1-sCMV-SaCas9-sgRNA_{UL54} or ssAAV1-sCMV-SaCas9-sgRNA_{UL30}. Analysis was performed at 10 days after AAV exposure. Mutagenic event detection by T7E1 assay in DNA from cultured TG neurons treated with either **b** $10^6$ vgs ssAAV1-sCMV-SaCas9-sgRNA_{UL54} or **c** ssAAV1-sCMV-SaCas9-sgRNA_{UL30}. The HSV regions containing the target site for each sgRNA were PCR amplified from total genomic DNA obtained from the right ipsilateral TG. Products were subjected to T7E1 digestion and separated on a 3% agarose gel. mw, molecular weight size ladder, V: no sgRNA, 13: sgRNA_{UL54-13}, 17: sgRNA_{UL54-17}, 26: sgRNA_{UL54-26}, 1: sgRNA_{UL30-1}, 2: sgRNA_{UL30-2}, 3: sgRNA_{UL30-3}, 6: sgRNA_{UL30-6}, 9: sgRNA_{UL30-9}, 10: sgRNA_{UL30-10}. Schematic representation of PCR amplicon with full-size and T7E1 cleavage product sizes indicative of HSV-specific Cas9 cleavage and mutagenesis is provided below each gel. The location of the sgRNA site in the PCR product is indicated by a black (efficient) or gray (inefficient) arrowhead, and resulting T7 digest products are indicated for efficient sgRNA. **d** Mice were latently infected with $2 \times 10^5$ PFU HSV-1(F), and 28 days later were injected in the right whisker pad with $10^{12}$ vgs ssAAV1-sCMV-SaCas9-sgRNA_{UL54} ($n = 5$ per sgRNA_{UL54}). Analysis was performed at either 28 ($n = 3$ mice per sgRNA_{UL54}) or 56 ($n = 2$ mice per sgRNA_{UL54}) days post AAV exposure. **e** Levels of HSV genomes were quantified by ddPCR in right (ipsilateral) TGs from infected mice. **f** Mutagenic event detection by NGS analysis of the PCR products used in the T7E1 analysis (See Supplementary Fig. 3b, c). sgRNA_{UL54-13} (circles), sgRNA_{UL54-17} (squares), and sgRNA_{UL54-26} (triangles). The gel images were cropped. All data are presented as mean values ± SD. bp: base pairs. Source data are provided as a Source data file.

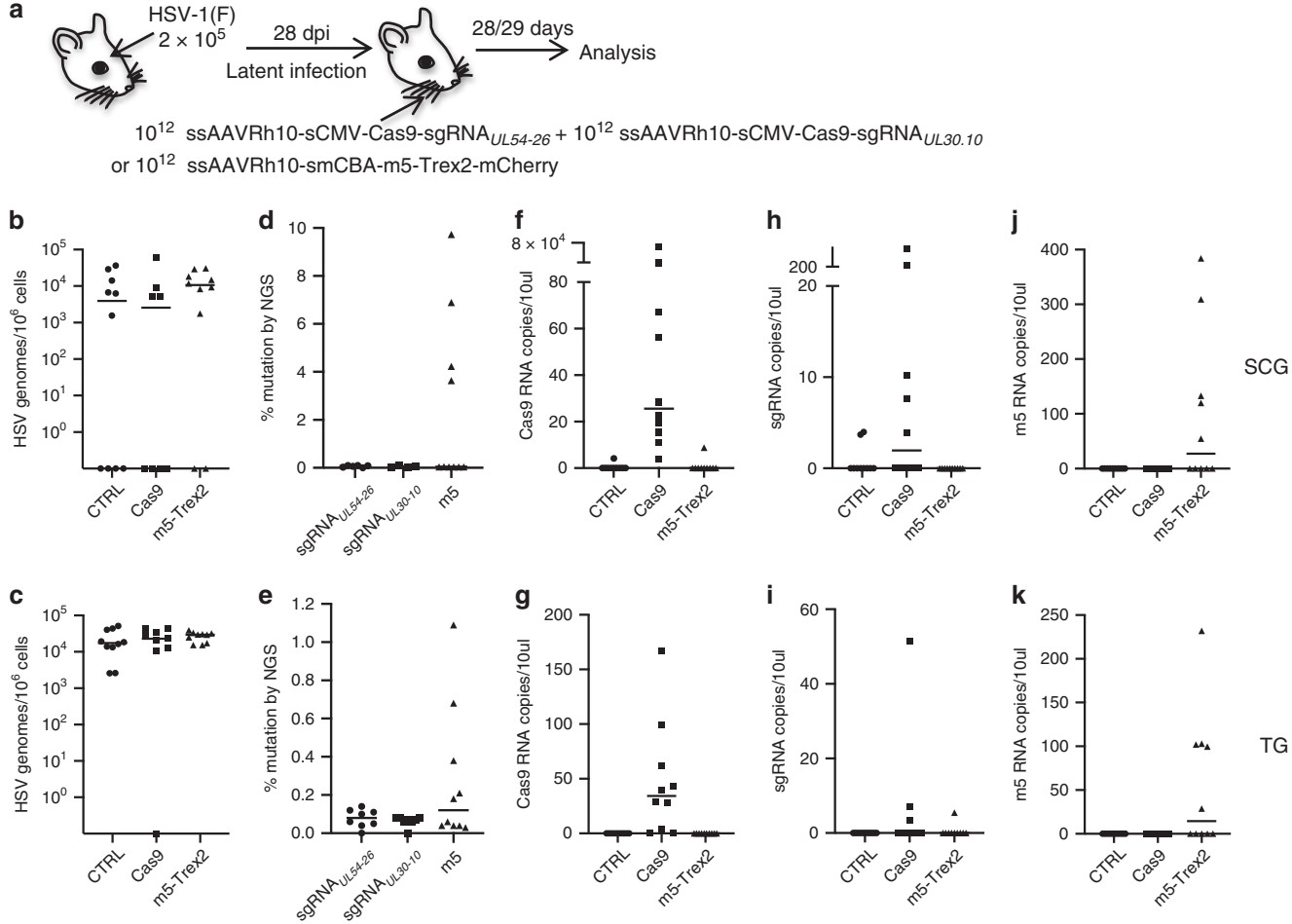

**Fig. 6 Dual sgRNA therapy did not increase SaCas9 gene editing efficiency of latent HSV. a** Mice were latently infected with $10^5$ PFU HSV-1(F), and 28 days later were left untreated (CTRL, $n = 10$ circles) or administered by RO injection either dual sgRNA therapy consisting of $10^{12}$ vgs ssAAVRh10-sCMV-SaCas9-sgRNA_{UL54-26} and $10^{12}$ vgs ssAAVRh10-sCMV-SaCas9-sgRNA_{UL30-10} (Cas9, $n = 10$ squares) or meganuclease therapy of $10^{12}$ vgs ssAAVRh10-smCBA-m5-Trex2-mCherry (m5 + Trex2, $n = 10$ triangles). At 28-29 days post AAV exposure **b**, **c** levels of HSV genomes were quantified by ddPCR in SCGs and right (ipsilateral) TGs from infected mice. **d**, **e** NGS analysis was performed to detect mutation at the site targeted by either sgRNA_{UL54-26}, sgRNA_{UL30-10} or m5 in latent HSV from SCG and TG of treated mice. **f**, **g** Detection of Cas9 mRNA by RT-qPCR in SCG (**f**) and TG (**g**) of infected mice. **h**, **i** Detection of sgRNA by RT-ddPCR in SCG (**h**) and TG (**i**) of infected mice. **j**, **k** Detection of m5 mRNA by RT-qPCR in SCG (**j**) and TG (**k**) of infected mice. All data are presented as mean values ± SD. Source data are provided as a Source data file.

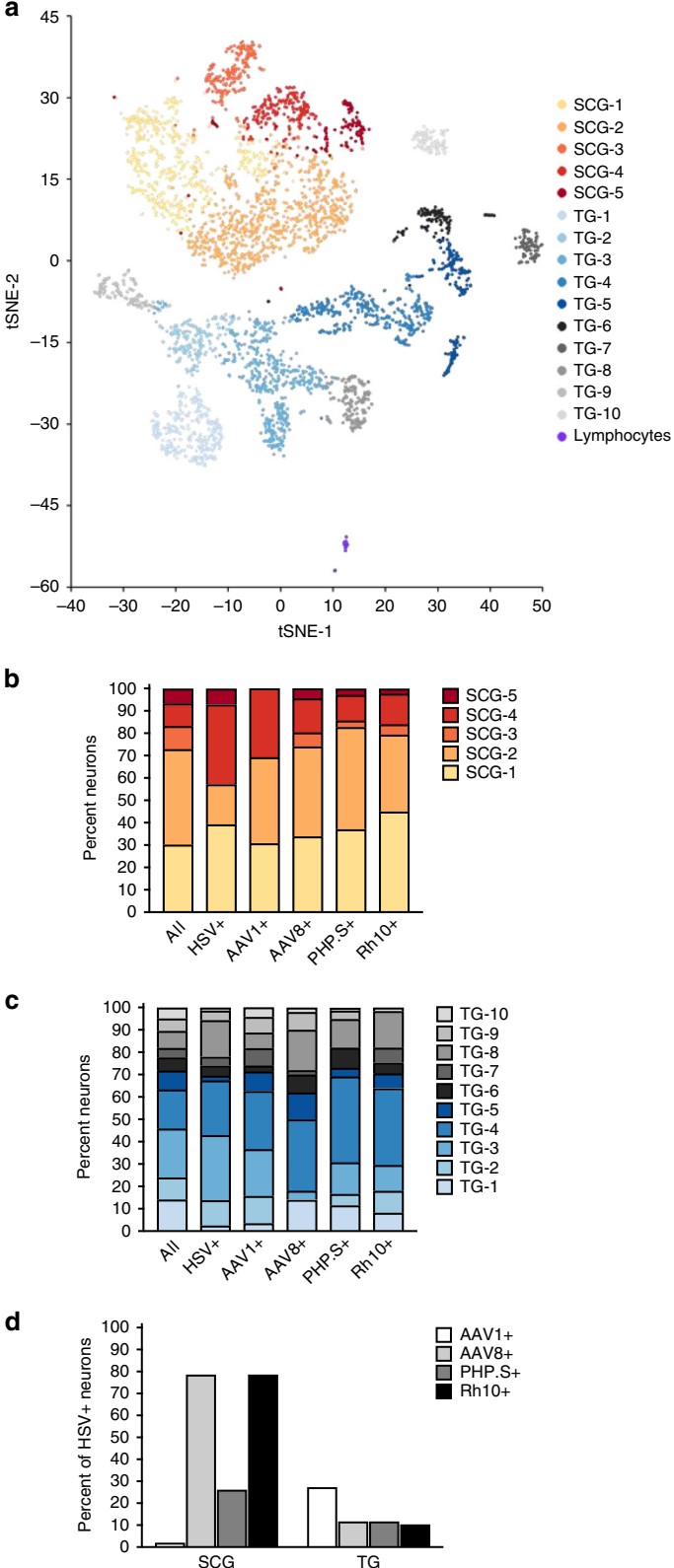

**Fig. 7 Single-cell RNA-seq analysis of purified neurons. a** tSNE plot of neurons dividing them into transcriptionally defined clusters. Clusters of the neurons purified from the SCG ($n = 2041$ with 94,797 mean reads and 5635 median genes per cell) colored with orange/red hues (SCG-1–5), and neurons purified from the TG ($n = 2319$ with 99,817 mean reads and 5908 median genes per cell) are colored with blue/gray hues (TG-1–10). Lymphocytes are purple. **b**, **c** Fractional distribution of each neuronal cluster within all neurons, or neurons expressing HSV RNA or the transgenes carried by the indicated AAV serotype, across the SCG and TG. **d** Calculated percentage of HSV+ cells in the SCG and TG that also express a given transgene mScarlet (AAV1+); mEGFP (AAV8+); DsRed.Express2 (PHP.S+); TagBFP2 (Rh10+).

Fig. 7b, Supplementary Figs. 8–9, Supplementary Tables 2–3). Within the TG, transgene expression from all serotypes was non-randomly distributed across the neuronal clusters ($\chi^2$: AAV1 $p = 0.0023$; AAV8 $p = 0.0002$; PHP.S $p < 0.0001$; Rh10 $p = 0.0006$; Fig. 7c, Supplementary Figs. 8–9, Supplementary Tables 2–3). 62% of AAV-Rh10+ neurons vs. 54% of AAV8+ neurons were collectively in TG-3, TG-4, and TG-8, the neuronal clusters which contained 70% of the HSV+ neurons, which may partially explain why Rh10 provided slightly higher levels of gene editing in the TG than did AAV8 (Supplementary Fig. 1). Our analysis did not identify specific cellular transcripts unique to neurons transduced by specific AAV serotypes

Finally, we evaluated the overlap of HSV gene expression compared with each AAV serotype across neuronal clusters. Strikingly, 79% of HSV+ neurons in the SCG were calculated to also express an AAV8 or Rh10 transgene, whereas only approximately 10% of HSV+ cells in the TG detectably expressed those transgenes (Fig. 7d, Supplementary Fig. 10, and Supplementary Table 4). These scRNA-seq data collectively demonstrate that the distribution of AAV to HSV-containing neurons is a crucial parameter in modulating DNA editing efficiency, and suggest that choosing AAV serotypes, promoters, and delivery methods to maximize the overlap of AAV transduction with HSV infection might increase the efficacy of meganuclease gene therapy against HSV.

**Combination of AAV serotypes leads to higher gene editing.** The results from the scRNA-seq analysis indicated that individual AAV serotypes vary in their tropism for specific neuronal subsets, suggesting that a combination of AAV serotypes could facilitate efficient ganglionic meganuclease delivery. Therefore, latently infected mice were treated with dual-meganuclease therapy using either single (AAV1, AAV8 or AAVRh10), double (AAV1 and AAV8, AAV1 and AAVRh10, or AAV8 and AAVRh10) or triple (AAV1, AAV8, and AAVRh10) AAV serotype combinations (Fig. 8a). Analysis of SCG and TG collected a month later showed that all AAV combinations, except for AAV1 alone, transduced the ganglia at similar levels (Fig. 8b, c). Consistent with the results above, dual-meganuclease therapy delivered using AAVRh10, either alone or in combination with 1 or 2 of the other AAV serotypes, led to the highest loss of latent viral genomes from SCG, with the greatest decrease in viral load obtained in the triple AAV combination (92%, Fig. 8d, Table 2). As predicted from our scRNA-seq data, the smallest decrease in HSV genomes was observed in animals receiving AAV1, unless AAVRh10 was included. Similarly, in the TG, the greatest loss of latent HSV genomes was detected in mice having received the dual-meganuclease therapy delivered using the triple AAV serotype combination (54.8%, Fig. 8e, Table 2). In agreement with our previous results, despite the efficient elimination of HSV genomes, gene editing in the residual HSV genomes was low, generally <10% for the HSV1m5 site (Fig. 8f, g) and ranging from undetectable to 8% for the HSVm8 site (Fig. 8h, i).

Taken together, these data suggest that in order to maximize HSV gene therapy in both SCG and TG, meganuclease delivery may benefit from a combination of different AAV serotypes to optimally target all HSV-infected neurons in both autonomic and sensory ganglia. Furthermore, the results showed that AAV1 may not be an ideal serotype to be used in combination with AAVRh10 and AAV8. Additional experimentation will be required to identify the optimal AAV serotype(s) to include in therapeutic combinations to maximize the meganuclease delivery to all HSV-infected ganglionic neurons.

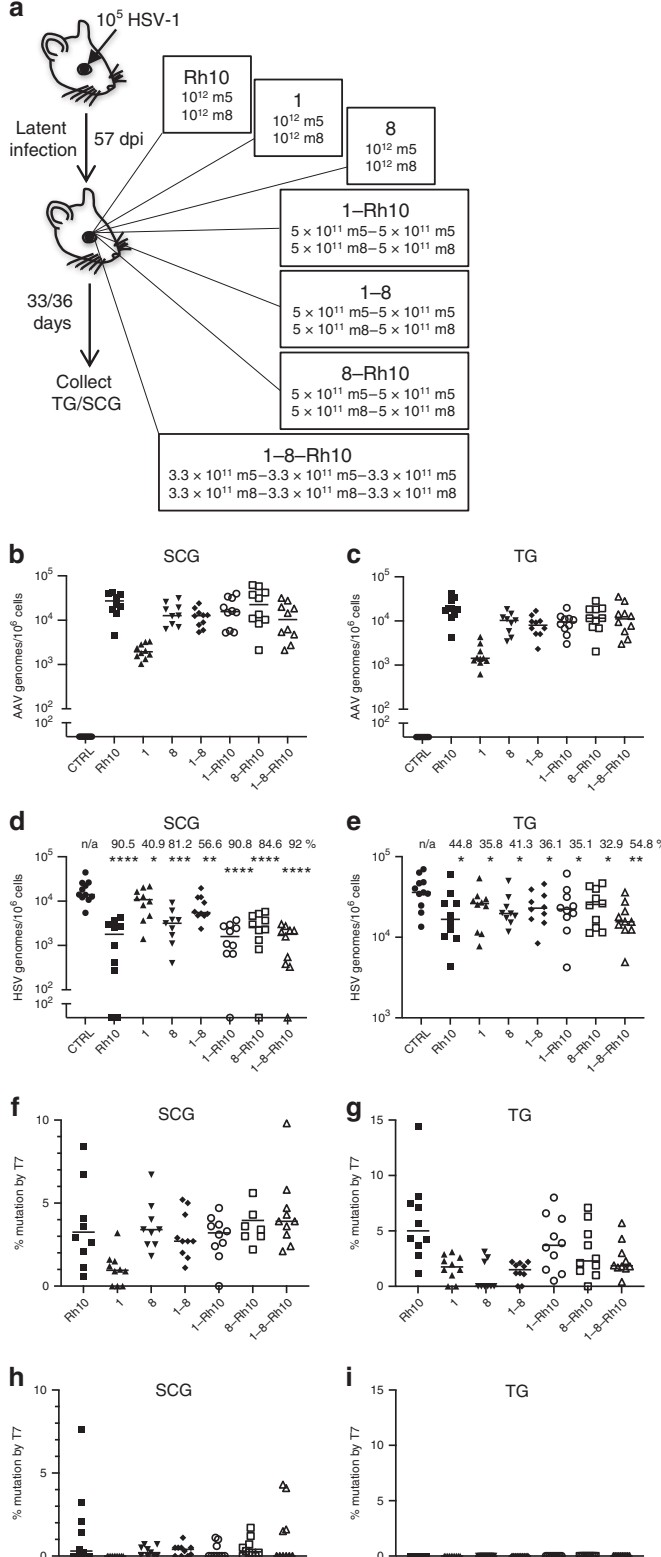

## Discussion

In the current work, we took advantage of a relatively simple mouse model of HSV infection to perform a set of iterative studies to increase the efficiency of AAV-delivered gene editing enzymes targeting HSV. We achieved a reduction in HSV genomes of >90% in SCG and >50% in TG of treated animals. This

**Fig. 8 AAV serotype combination for the delivery of dual-meganuclease therapy. a** Mice latently infected with $10^5$ PFU HSV-1 17+ for 57 days, were either left untreated (Controls, CTRL, $n = 10$) or administered $2 \times 10^{12}$ vgs total of scAAV-CBh-meganuclease combination by RO injection to deliver meganuclease dual therapy (m5 + m8, $n = 10$ per AAV combination). Mice received either single (Rh10, 1, or 8), dual (1–8, 1-Rh10 or 8-Rh10) or triple (1-8-Rh10) AAV serotype combination. At 33–36 days post meganuclease-therapy, SCGs and right (ipsilateral) TG were harvested, and **b**, **c** AAV genomes and **d**, **e** HSV genomes from SCG (**b-d**) and TG (**c-e**) from infected mice were quantified by ddPCR. Percentages of HSV genomes decrease in ganglia from dual-meganuclease-treated mice compare with control untreated (CTRL) mice are indicated for each AAV combination (**d**, **e**). n/a: not applicable. **f–i** Gene editing at the m5 target (**f**, **g**) site and m8 target site (**h**, **i**) were quantified by T7E1 assay in SCG (**f-h**) or TG (**g-i**). ns: not significantly different from controls, \*$p < 0.05$, \*\*$p < 0.01$, \*\*\* $p < 0.001$, \*\*\*\* $p < 0.0001$ significantly different from controls. All data are presented as mean values ± SD. Statistical analysis was conducted using one-tail, unpaired $t$-test. Exact $p$ values are indicated in Table 2. Source data are provided as a Source data file.

represents a dramatic improvement upon our previous report in which we observed a maximum of about 4% gene editing with no loss of viral genomes. While HSV does not reactivate spontaneously from ganglia of living mice, the virus does reactivate from mouse neurons after explantation, and our results demonstrate 95% (SCG) to 55% (TG) reduction in viral genomes produced de novo in ganglionic explants after meganuclease treatment of latently infected mice. This is consistent with previous work demonstrating that ganglionic HSV load is a major determinant of the frequency of viral reactivation[31–34]. If translated to humans, such an outcome could be useful in reducing the likelihood of viral reactivation, shedding, and transmission to others[35]. We had previously hypothesized that linearization of episomal latent HSV genomes by gene-editing enzymes might lead to their degradation and loss if DNA repair and recircularization were unsuccessful[13]. However, the gene-editing frequencies achieved in our earlier work following exposure to a single meganuclease were insufficient to generate enough HSV genome loss to be detected, even by precise ddPCR assays[13]. In contrast, the robust gene editing achieved in the current report after dual-meganuclease therapy led to loss of HSV genomes to a degree readily detectable by ddPCR. Of the remaining viral genomes, an average of 4–6% were mutated, which likely further contributes to suppressing HSV reactivation.

One limitation to our study is that we cannot distinguish whether the observed reduction in de novo production of viral genomes resulted from a reduction in the number of reactivation events, or instead from a decrease in virus production after a reactivation event has initiated. Our previous results suggest that

newly synthesized HSV genomes are efficiently targeted by nucleases[13], while results from our group and others demonstrate that chromatin modification of latent HSV can reduce the efficacy of gene editing[13,20,25,36]. This has important implications for the ultimate embodiment of gene editing for HSV, as it will determine whether sustained long-term expression of nuclease (as achieved using AAV vectors[22]) is required for therapeutic benefit, or whether alternative transient delivery approaches might be sufficient. On the other hand, the substantial improvement in reduction of latent HSV load reported here (>90% reduction compared to no detectable reduction in our previous report) warrant cautious optimism that with further optimization, the ganglionic HSV load might be reduced to levels at which sustained nuclease expression is unnecessary. Ongoing experiments in mice and other model systems will shed additional light on this issue.

Our findings convincingly demonstrate the value of single-cell analysis for optimizing in vivo gene therapy. For gene editing against persistent viruses to have maximum efficacy, transgene must be efficiently delivered to cells containing virus. Our results demonstrate that a substantial fraction of HSV-expressing cells also express detectable reporter transgene indicating delivery by AAV, particularly in the SCG where editing was highest. Furthermore, our results clearly demonstrate that distinct neuronal subsets are preferentially transduced by different AAV serotypes. For example, AAV1 transduced 17.4% of cells in TG-3, 16.6% of cells in TG-8, but only 2.3% of cells across the whole SCG, none of which were HSV+, while AAVRh10 transduced only 5.1% of cells in TG-3, but 20.7% of cells in TG-8 and 23.5% of cells across the SCG including 78.6% of the HSV expressing cells. Together these results suggested that combinations of AAV serotypes may be required to efficiently target all neurons containing latent HSV, which we tested in a follow-up experiment where rationally-selected AAV serotype combinations were used for the delivery of dual-meganuclease therapy. As predicted, the triple AAV serotype combination led to the greatest decrease in HSV loads, 92% for the SCG and 54.8% for the TG. Our results suggest that the AAV serotypes combined for the delivery of meganuclease therapy need to be carefully chosen, and likely can be further improved. For example, the use of AAV1 alone resulted in the lowest decrease in HSV loads in both SCG and TG, and the addition of AAV1 appeared to add little efficacy to AAVRh10 or AAV8, either alone or in combination. Future work should focus on identifying the optimal combinations of AAV to ensure full coverage of all infected neurons, and to establish whether such combinations are also effective in other model systems, which would facilitate human translation of this approach.

Somewhat surprisingly, in our system meganucleases provided substantially higher gene editing than did Cas9 with any of the tested gRNAs. The simplest explanation for this may be due to relative expression levels; due to its larger size Cas9 requires

**Table 2 HSV loads after delivery of dual-meganuclease therapy using AAV combinations.**

| Groups ($n = 9 - 11$ mice) | Mean viral load per $1 \times 10^6$ cells SCG/TG | Percent reduction SCG/TG % | $p$ values SCG/TG |
|---|---|---|---|
| CTRL | $1.87 \times 10^4 / 2.17 \times 10^4$ | n/a | n/a |
| Rh10 | $1.78 \times 10^3 / 2.52 \times 10^4$ | 90.5/44.8 | $<10^{-4}/0.013$ |
| 1 | $1.10 \times 10^4 / 2.30 \times 10^4$ | 40.9/35.8 | 0.034/0.026 |
| 8 | $3.51 \times 10^3 / 2.51 \times 10^4$ | 81.2/41.3 | 0.0003//0.013 |
| 1–8 | $8.14 \times 10^3 / 2.55 \times 10^4$ | 56.6/36.1 | 0.005/0.02 |
| 1-Rh10 | $1.73 \times 10^3 / 2.64 \times 10^4$ | 90.8/35.1 | $<10^{-4}/0.036$ |
| 8-Rh10 | $2.89 \ 10^3 / 2.17 \ 10^4$ | 84.6/32.9 | $<10^{-4}/0.036$ |
| 1–8-Rh10 | $1.49 \times 10^3 / 1.78 \times 10^4$ | 92.0/54.8 | $<10^{-4}/0.001$ |

All data are presented as mean values and statistical analysis was conducted using one-tail, unpaired $t$-test. Source data are provided as a Source Data file.
*CTRL* Control n/a not applicable.

delivery by single-stranded (ss)AAV vectors, while meganucleases fit easily into the more transcriptionally efficient self-complementary (sc)AAVs, which do not require de novo second strand synthesis or intermolecular annealing for transgene expression. It has been demonstrated previously that the use of scAAV vectors rather than ssAAV greatly enhances transduction efficiency[37]. Size restrictions also limited the choice of potential promoters for Cas9 expression. On the other hand, we were able to achieve high levels of HSV gene editing in vitro using these same AAV/promoter/Cas9 constructs, so if true, these factors must be more important in vivo than in vitro. The low expression of sgRNA detected in SCG and TG (only 40 and 20%, respectively) may partially explained the poor gene editing in vivo. However, it is not the sole explanation, since m5 mRNA was detected in only 50% of the ganglia, yet the observed mutation reached up to 9.9% or 1.1% in SCG and TG, respectively. We also cannot rule out the possibility that our in vitro model systems may not fully recapitulate the state of viral latency achieved in vivo, particularly in regard to viral chromatinization (reviewed in ref. [38]). However, we have previously shown no detectable differences in the frequency of mutation induced by HSV-specific meganucleases in neuronal cultures established from TG of mice during acute vs. latent infection, which should differ in chromatinization. An intriguing possibility is that meganucleases might target highly compact and heterochromatinized viral genomes better than Cas9, which would be consistent with their evolution in eukaryotes, compared to the evolution of Cas9 in prokaryotes. Future studies should systematically evaluate expression of various classes of gene editing enzymes, and their efficacy against specific genomic and viral targets, to address this issue.

Taken together, our results provide strong support for the continued development of gene editing as a strategy against latent HSV infections, as well as other chronic infections such as HBV and HIV. A recent study using a similar approach for HIV shows eradication of detectable HIV in some (but not all) humanized mice[39]. We have performed similar studies in HBV-infected, liver-humanized mice, and achieved a near one-log reduction in intrahepatic HBV cccDNA levels, along with a remarkable improvement in human hepatocyte survival[40]. Although the genetic diversity of these viruses remains a concern, careful informatics analysis can effectively address this issue[41]. Moving this approach toward clinical application will require careful examination of its safety, including confirming the absence of off-target cleavage within the genome. The results also suggest important potential advantages for meganucleases, an under-appreciated class of gene editing enzymes, in particular their compact nature that allows utilization of highly-optimized scAAV vectors and a wider selection of promoters. Although sequence-specific meganucleases are more difficult to develop compared with CRISPR/Cas9, this represents only a minor issue for targets such as HSV, which have limited genetic diversity and slow rates of genomic evolution. In principle, a handful of optimized meganucleases should be sufficient to cover the full diversity of HSV-1 and HSV-2 observed in human infection, in contrast to other less conserved viruses, such as HBV and HIV[41,42]. The levels of efficacy observed to date (>90% HSV reduction in SCG, and >50% in TG), if translated to humans, would be likely to meaningfully reduce HSV reactivation, shedding, transmission, and lesions. Further optimization of enzyme delivery and the meganucleases themselves are likely possible, and thus a cure for HSV infection may ultimately be within reach.

## Methods

**Cells and herpesviruses.** HEK293[43] and Vero cell lines (ATCC # CCL-81) were propagated in Dubelcco's modified Eagle medium supplemented with 10% fetal bovine serum. HSV-1 strain F (kindly provided by Dr. J. Blaho) or syn17+ (kindly

provided by Dr N. Sawtell) were used for the experiments and were propagated and titered on Vero cells.

**AAV production and titering.** The following AAV vector plasmids were used to generate the AAV stocks in this study: pscAAV-CBh-m5, pscAAV-CBh-m8, pscAAV-CBh-m4, pssAAV-smCBA-m5-T2A-Trex2-2A-mCherry, pssAAV-sCMV-SaCas9-U6-sgRNA, pssAAV-CMV-SaCas9-U6-sgRNA, pssAAV-nEF-SaCas9-U6-sgRNA, pscAAV-CBh-NLS-mScarlet, pscAAV-CBh-NLS-mEGFP, pscAAV-CBh-NLS-DsRed-Express2, and pscAAV-CBh-NLS-mTagBFP2. AAV stocks of all serotypes were generated by transiently transfecting 293 cells using PEI at a ratio of 4:1 (μl PEI:μg DNA) according to the method of Choi et al.[44]. Briefly, $1.6 \times 10^7$ HEK293 cells were transfected with 28 μg DNA comprised of the DNA for a scAAV or ssAAV vector plasmid, a plasmid that expresses the AAV rep and capsid proteins, and a helper plasmid that expresses adenovirus helper proteins (pHelper) at the ratio of 5:1:3, respectively. At 24 h post-transfection media was changed to serum-free DMEM and after 72 h cells were collected and re-suspended in AAV lysis buffer (50 mM Tris, 150 mM NaCl, pH 8.5) before freeze-thawing 4 times. AAV stocks were purified by iodixanol gradient separation[44,45] followed by concentration into PBS using an Amicon Ultra-15 column (EMD Millipore) and stored at −80 °C. All AAV vector stocks were quantified by qPCR using primers/probe against the AAV ITR, with linearized plasmid DNA as a standard, according to the method of Aurnhammer et al.[46]. AAV stocks were treated with DNase I and Proteinase K prior to quantification.

**Establishment of neuronal cultures.** Neuronal cultures were established from TG harvested from mice at day 7 post infection with $2 \times 10^5$ PFU HSV-1(F) onto scarified cornea[13]. Briefly, neuronal cultures were established after enzymatic digest with collagenase and dispase (Invitrogen, Carlsbad, CA)[47] and purification of the resulting cell homogenates using a percoll gradient (12.5 and 28%)[48]. Neurons were counted and plated on poly-D-lysine- and laminin-coated 12 mm round slides (BD Biosciences, San Jose CA) at a density of 4000 neurons per well. Neurons were cultured without removing the non-neuronal cells that provide important growth support, and therefore these cultures contained a mixed population of neurons, satellite glial cells, and other cell types. Cultures were maintained with complete neuronal medium, consisting of Neurobasal A medium supplemented with 2% B27 supplement, 1% PenStrep, L-glutamine (500 μM), and nerve growth factor (NGF; 50 ng/ml). Medium was replaced every 2–3 days with fresh medium. Acyclovir (100 nM, Sigma) was added to the culture medium for the first 5 days.

**HSV infection and AAV inoculation of mice.** Mice were housed in accordance with the institutional and NIH guidelines on the care and use of animals in research. 6–8 week old female Swiss-Webster mice (Charles River) were used for all studies. For ocular HSV infection mice anesthetized by intraperitoneal injection of ketamine (100 mg per kg) and xylazine (12 mg per kg) were infected with $2 \times 10^5$ PFU of HSV1(F) or syn17+ following corneal scarification of the right eye using a 28-gauge needle. For AAV inoculation, mice anesthetized with ketamine/xylazine were unilaterally administered the indicated AAV vector dose by either intradermal whisker pad (WP) injection, retro-orbital (RO) or tail-vein (TV) injection. The right (ipsilateral) TG and both SCGs were collected at the indicated time. Presence of AAV in ganglia of treated mice was confirmed by ddPCR (Supplementary Fig. 4).

**Tissue explant reactivation.** HSV was reactivated by incubating collected TG and SCG in 10% FBS-DMEM culture medium for 24 h, followed by total genomic DNA extraction as described below. After tissue reactivation, a statistically significant increase of 2 to 3-fold in HSV genomes is detected in reactivated compare to unreactivated (latent) tissues in untreated (control mice) (Fig. 2g, h, Supplementary Fig. 2f).

**HSV target site PCR amplification.** Total genomic DNA (gDNA) was extracted using either the DNeasy Tissue & Blood micro kit (Qiagen, Valencia, CA) for neuronal cultures or the DNeasy Tissue & Blood mini kit (Qiagen, Valencia, CA) for whole TGs. Platinum *Pfx* DNA polymerase (Invitrogen, Carlsbad, CA) and 5 ul of gDNA were used to PCR amplify the region containing the target site for either HSV1m5 with U_L19 primers: forward 5′-CTGGCCGTGGTCGTACATGA and reverse 5′-TCACCGACATGGGCAACCTT, HSV1m8 with UL30 primers: forward 5′-GAGAACGTGGAGCACGCGTACGGC and reverse 5′-GGCCCGGTTTGAGACGGTACCAGC, HSV1m4 with ICP0 primers: forward 5′-GACAGCACGGACACGGAACT and reverse 5′-TCGTCCAGGTCGTCGTCATC,

*Sa*Cas9/sgRNA_UL54 (sgRNA13, sgRNA17 and sgRNA26) with U_L54 primers: forward 5′-GACCGCATCAGCGAGAGCTT and reverse 5′-CTCGCAGACACGACTCGAAC, or *Sa*Cas9/sgRNA_UL30 (sgRNA1, and sgRNA10) with U_L30-primers: forward 5′-CGGCCATCAAGAAGTACGAG and reverse 5′-AAGTGGCTCTGGCCTATGTC, with thermocycler conditions of: 94 °C 5 min, 40–45 cycles (94 °C 30 s, 60 °C 30 s, 70 °C 30 s), and then 70 °C 5 min.

**T7 endonuclease 1 (T7E1) assay.** The T7 endonuclease assay and quantification to determine the levels of gene disruption were performed as follows. After PCR

amplification of target site from HSV genomes, followed by purification using Zymo Research clean and concentrator-5 kit (Zymo Research, Irvine CA), 300 ng of DNA amplicon was denatured for 10 min at 95 °C and slowly reannealed by cooling down to room temperature. DNA was then digested with 5–10 units of T7 endonuclease (New England Biolabs,) for 30–60 min at 37 °C and resolved in an agarose gel. Quantification of gene disruption was performed using ImageJ software (NIH[49]) and calculated using the formula: $100 \times (1 - [1 - \text{fraction cleaved}]1/2)$ where fraction cleaved = density of cleaved product/(density of cleaved product + density of uncleaved product).

**ddPCR quantification**. Viral genome quantification by ddPCR was performed using an AAV ITR primer/probe set, and a gB primer/probe set for HSV as described previously[13]. Cell numbers in tissues were quantified by ddPCR using mouse-specific RPP30 primer/probe set: For 5′-GGCGTTCGCAGATTTGGA, Rev 5′-TCCCAGGTGAGCAGCAGTCT, probe 5′-ACCTGAAGGCTCTGCGCGGA CTC. In some control ganglia, sporadic samples showed positivity for AAV genomes, although at levels typically >2–3 logs lower than ganglia from treated mice that had received AAV. We attribute this to low-level contamination of occasional tissue samples.

**Illumina next-generation sequencing (NGS)**. Next-generation sequencing of meganuclease target sites was performed using PCR products generated with the target site-specific primers described above and a MiSeq sequencer (Illumina)[13].

**Single cell RNA analysis**. Swiss-Webster mice were latently infected with $10^5$ PFU HSV-1 syn 17+ via the ocular route, and after 60 days injected with one of 4 different AAV serotypes: 1, 8, PHP.S, and Rh10, each carrying a unique fluorescent protein transgene: mScarlet, mEGFP, DsRed.Express2, and TagBFP2, respectively under the CBh promoter. For each serotype three mice were independently injected with $10^{12}$ AAV genomes subcutaneously in the whisker pad (AAV1) or intravenously in the retro-orbital vein (AAV8, PHP.S, and Rh10). Three weeks later, TG and SCG from animals were collected and each tissue (TG or SCG) was pooled from all animals for neuron isolation via enzymatic tissue digest (see above), followed by density gradient centrifugation and enrichment using the Neuron Isolation Kit (Miltenyi BioTech.), which allows untouched neurons to flow through the column while non-neuronal cells remain bound.

Only 2 AAV8-mEGFP animals were used for cell preparations or analyses. Tissue and isolated neurons were maintained in ice-cold Neurobasal A medium supplemented with 2% B27 supplement, 1% PenStrep, L-glutamine (500 μM) or PBS throughout the procedure except during the enzymatic tissue digestion steps. Cells were encapsulated and scRNA-seq libraries were prepared in the Genomics Core Facility at the FHCRC using the Chromium Single Cell 3′ Library and Gel Bead Kit v2 from 10X Genomics according to manufacturer instructions. 10× Genomics Single Cell 3′ expression libraries were sequenced on an Illumina HiSeq 2500 running in High-Output mode with a paired-end (26 bp × 8 bp × 98 bp) sequencing strategy. The SCG and TG libraries were pooled and distributed over 8 sequencing lanes. Image analysis and base calling were performed using RTA Version 1.18.66.3. Sequencing reads were processed with the 10X Genomics 'Cell Ranger' v2.1.0 and with Seurat v2.3.4[50]. We obtained high quality sequence data from 2372 purified TG neurons and 2,172 SCG neurons.

**Statistics and reproducibility**. GraphPad Prism 7 software was used for statistical analyses. Comparison of HSV loads was performed with multiple t-test, with alpha = 0.05%. Each tissue was analyzed individually, without assuming a consistent SD. HSV and AAV distributions in scRNA-seq experiments were analyzed using the $\chi^2$ test with alpha = 0.05. Because tissue from animals injected with the different AAV serotype/transgene were pooled, cell counts were normalized to input by multiplying cell count by number of animals receiving that serotype divided by total number of animals. The mean and error bars representing the standard deviation are shown on each graph. The findings of the studies were reproduced across the experiments using different experimental set-ups.

**RT-ddPCR quantification of SaCas9, sgRNA, HSV1m5 expression**. AllPrep DNA/RNA kit (Qiagen, Valencia, CA) was used to isolate DNA and RNA from ganglia collected in the experiment presented in Fig.4. SaCas9, sgRNA and HSV1m5 expression was quantified with One-step RT-ddPCR kit (BIO-RAD, Hercules, CA) using 2 μl of RNA and the following primers/probe set: SaCas9 specific primers: SaCas9 forward 5′-CCGCCCGGAAAGAGATTATT, reverse 5′-CGGAGTTCAGATTGGTCAGTT, and probe [FAM]AGCTGCTG GATCAGATTGCCAAGA[MGB]; Tracr specific primers: TRACR LS forward 5′-TGCCGTGTTTATCTCGTCAACT, reverse 5′-CCCGCCATGCTACTTATCTA CTTAA, and probe [FAM]TTGGCGAGATTTTT[MGB]; HSV1m5 specific primers: m5 mega forward 5′-TGGACAGCCTGAGCGAGAA, reverse 5′-GCAGA GACAGAGGAGCAATGTG, and probe [FAM]CGGCCGGTGATTCCTC TGTTTCTAATTC[BHQ]. The cycling steps were as follows: reverse transcription 50 °C, 60 min, enzyme activation 95 °C, 10 min, 40 cycles (95 °C, 30 s, 60 °C, 1 min, and 70 °C, 30 s), and then enzyme deactivation 98 °C, 10 min.

**Study approval**. All animal procedures were approved by the Institutional Animal Care and Use Committee of the Fred Hutchinson Cancer Research Center.

**Reporting summary**. Further information on research design is available in the Nature Research Reporting Summary linked to this article.

## Data availability
Single-cell RNA sequencing data is available via GEO under accession GSE151811. Raw sequence data has been uploaded to SRA under BioProject PRJNA330548. Meganuclease sequences are Cellectis proprietary information; material and information requests can be made to R. Galetto (roman.galetto@cellectis.com) and P. Duchateau (philippe. duchateau@cellectis.com). Source data are provided with this paper.

## Code availability
Code used to process NGS data to determine the mutation rates is available at [https://github.com/proychou/TargetedMutagenesis], and code used for the scRNAseq analysis is available at [https://github.com/proychou/HSV_10x].

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

## Acknowledgements

This research was supported by NIH grants R21AI117519 and R01AI132599 (Jerome), the Caladan Foundation, and the Genomics, Experimental Histopathology and Comparative Medicine Shared Resources of the Fred Hutch/University of Washington Cancer Consortium (P30 CA015704). We thank Lawrence Corey and Ashley Sherrid for critical reading of the paper.

## Author contributions

M.A., D.E.S., D.S., P.R., M.L.H., and K.R.J. were involved in the conception and design of the study. M.A., M.A.L., A.K.H., L.M.K., D.E.S., L.S., M.L.H., N.M., A.T., A.G., H.S.D.S.F., D.S., and K.R.J. performed, oversaw, interpreted, and/or generated data. More specifically, M.A. was involved in all the data generation, analysis, and interpretation of all aspects of the study. M.A., M.A.L., A.K.H., and D.E.S. performed the animal studies. M.A., N.M., and A.T. generated the NGS data sets, A.G. provided supervision of the NGS, and P.R. performed the computational analysis of the NGS data sets. L.S. performed all the ddPCR quantification. M.A., M.L., A.K.H., D.E.S., and H.S.D.S.F. generated the AAV stocks used in this study. M.A., M.A., and M.L. performed experiments necessary for the scRNAseq, and P.R. and D.E.S. performed the computational analysis of the data. M.A. and K.R.J. wrote the paper. R.G. and P.D. provided plasmids with the endonuclease genes. All authors read and edited the paper.

## Competing interests

R.G. and P.D. are employees of Cellectis SA. All the other authors declare no competing interests.
