## [Peer Review File · Nature Communications]

Gene editing and elimination of latent herpes simplex virus in vivoEditorial Note: This manuscript has been previously reviewed at another journal that is not operating a transparent peer review scheme. This document only contains reviewer comments and rebuttal letters for versions considered at *Nature Communications*.

REVIEWERS' COMMENTS:

Reviewer #2 (Remarks to the Author):

Aubert et al submit a manuscript in which there is considerable optimization of the AAV mediated delivery of virus specific meganucleases (and Cas9) to try and target the latent HSV genome in vivo, using a murine model of ocular infection and latency. The goal is to remove latent genomes or at least render them unable to reactivate. This goal is enormously important because at present the latent state is quite refractile to any kind of antiviral therapy because of its largely gene silent state. Reactivation is the cause of considerable HSV morbidity, including blinding ocular disease. While this work is not entirely new and out of the blue, the authors correctly claim a vast improvement on previous work. In that study they were only able to mutate 4% of latent genomes in vivo, itself a huge step forward, although they were not able to reduce the genomic load. Here, using a combination of optimized parameters, they can achieve much higher rates of mutation and show genome removal from the superior cervical ganglia at up to 90%, and remove about half of the genomes in the trigeminal ganglia. This is under-represented in the figures because the authors use log graphs, but it is a major step forward.

The authors evaluate the use of meganucleases singly and in combinations to obtain multiple rather than single cleavage events, which work better. They also evaluate different AAV serotypes for delivery and us alone and in combination. There is a comparison of meganucleases with Cas9, although the claim of meganucleases working better than cas9 is not as solid. The overall conclusion is that a multi serotype with multi cleavage events would likely work best.

The authors also use a novel technique employing single cell sequencing to evaluate those neurons transduced by AAV and if this correlates with those hosting latent HSV genomes (although a caveat is that the latter are detected by expressing HSV LATS (a rather controversial area in the field is whether all latently infected neurons express LAT, but I do not think it needed to push that point here). This work should appeal to a wider audience and is a big step in the right direction. Yes, it is not known how much the work will extend to treatments of HSV disease, since it is likely that a different set of parameters, AAV serotypes, etc will be needed. The following issues should be addressed to improve the clarity of the manuscript.

The detail provided in the manuscript is sufficient for the work to be done by others. Indeed this article represents a very large amount of work.

1. I will be honest and state that the paper was a rather difficult read. It was not always presented in an easily readable manner and was often not straightforward to follow. Part of the problem is the continual switching between extended data and main data with rather minimal textual description of what the results meant and what the data in extended adds to the work. Some of the extended data has many multiple components, and is referred to with only a single or few lines of text. This made the results a bit laborious to go through.

2. It is not clear if the sgRNA and targets of the meganuclease were optimal, and clearly these have an influence on genome cleavage. For example, M8 and M5 clearly differ greatly in their cleavage

rates, yet both are single cleavers.

3. The comparison of a single meganuclease delivery to /CRSPR cas9 is still problematic; while they now compare cleavage using both ssAAV and the same serotype, the m5 is delivered with a TREX exonuclease while the CRSPR is not. What happens if a TREX is coexpressed with Cas9? For the most part TREX is not described in the text and should be. Also, the nature of the sgRNA guides might not be optimal, much as the meganucleases sometimes do not yield a high level of cleavage (m8 for example is a weak cleaver compared to M5). I do not understand why this point is still emphasized in the work. As they acknowledge there could be several variables that contribute, such as promoter, translation efficiency, sgRNA guide being optimal, and being able to access the presumably chromatinized DNA less efficiently than meganucleases.

4. The abstract seems to contradict. Line 2 states the source of the recurrent disease is the peripheral nervous system, but line 21 the SCG is part of the autonomic system, no? this needs a correction.

5. The use of scAAV and its likely importance in enhancing efficiency of transduction is vastly underplayed in this work.

6. A rather annoying aspect is that the authors use of names (eg AAV CBh rather than denoting the serotype of the AAV as Rh10). This added to the difficulty in following the studies.

7. Can the authors group the transduced neurons into a more descriptive terminology that would be familiar to the reader? Can they state if neurons groups transduced by specific AAV for the single cell analyses express certain markers (Trpv1, CGRP, NF200, etc)

8. Line 151 and 63 I think it would be pertinent o add a statement that SaCas9 is large and does not permit SCAAV to be developed

9. It is still not clear to me if reactivation is prevented or that virus emerging from reactivation is growth inhibited. The rebuttal explanation did not address this issue with clarity

10. A point of discussion that they should address is the different routes of virus administration. The HSV goes through the eye corneal surface, yet the AAV are given via a different route. Otherwise the discussion was thoughtful, well done and nicely provoking.

REVIEWERS' COMMENTS:

Reviewer #2 (Remarks to the Author):

Aubert et al submit a manuscript in which there is considerable optimization of the AAV mediated delivery of virus specific meganucleases (and Cas9) to try and target the latent HSV genome in vivo, using a murine model of ocular infection and latency. The goal is to remove latent genomes or at least render them unable to reactivate. This goal is enormously important because at present the latent state is quite refractile to any kind of antiviral therapy because of its largely gene silent state. Reactivation is the cause of considerable HSV morbidity, including blinding ocular disease. While this work is not entirely new and out of the blue, the authors correctly claim a vast improvement on previous work. In that study they were only able to mutate 4% of latent genomes in vivo, itself a huge step forward, although they were not able to reduce the genomic load. Here, using a combination of optimized parameters, they can achieve much higher rates of mutation and show genome removal from the superior cervical ganglia at up to 90%, and remove about half of the genomes in the trigeminal ganglia. This is under-represented in the figures because the authors use log graphs, but it is a major step forward.

The authors evaluate the use of meganucleases singly and in combinations to obtain multiple rather than single cleavage events, which work better. They also evaluate different AAV serotypes for delivery and us alone and in combination. There is a comparison of meganucleases with Cas9, although the claim of meganucleases working better than cas9 is not as solid. The overall conclusion is that a multi serotype with multi cleavage events would likely work best.

The authors also use a novel technique employing single cell sequencing to evaluate those neurons transduced by AAV and if this correlates with those hosting latent HSV genomes (although a caveat is that the latter are detected by expressing HSV LATs (a rather controversial area in the field is whether all latently infected neurons express LAT, but I do not think it needed to push that point here). This work should appeal to a wider audience and is a big step in the right direction. Yes, it is not known how much the work will extend to treatments of HSV disease, since it is likely that a different set of parameters, AAV serotypes, etc will be needed. The following issues should be addressed to improve the clarity of the manuscript.

The detail provided in the manuscript is sufficient for the work to be done by others. Indeed this article represents a very large amount of work.

1. I will be honest and state that the paper was a rather difficult read. It was not always presented in an easily readable manner and was often not straightforward to follow. Part of the problem is the continual switching between extended data and main data with rather minimal textual description of what the results meant and what the data in extended adds to the work. Some of the extended data has many multiple components, and is referred to with only a single or few lines of text. This made the results a bit laborious to go through.

To improve readability of the paper we have made several modifications to limit the need to switch between the main data and extended data, added textual description of the meaning of the results, and clarified what the data in extended data figures adds to the work.

- We have inserted the following statements or added to our statements:
 - o Line 100-102: "This serotype had demonstrated efficient transgene delivery in optimization studies, which showed that AAV-Rh10 serotype administered via retro-orbital injection led to the highest levels of HSV gene editing in ganglia (Extended data Fig. 1)."
 - o We Line 173-175: "We have shown that ganglionic explant reactivation resulted in an approximate two to three-fold increase in total HSV levels over fresh ganglia that can be measured by ddPCR (Extended data Fig. 2a-b)."
- We have split some of the main figures into several figures:
 - o Figure 1 was split into Fig. 1 and Fig. 2.

- Figure 2 was split into Fig. 3 and Fig. 4.
- Supplemental Figure 2 was split into Supplemental Figure 2 and Supplemental Figure 4.
- The data from supplementary Figure 2 showing the AAV levels in ganglia of treated mice in all the experiments presented in the main figures are now displayed in Supplemental Figure 4, rather than in individual figures in the main text. To reflect this change, the following sentence has been added in the Methods section, in the paragraph describing HSV infection and AAV inoculation of mice: “Presence of AAV in ganglia of treated mice was confirmed by ddPCR (Supplementary Fig. 4).”.

2. It is not clear if the sgRNA and targets of the meganuclease were optimal, and clearly these have an influence on genome cleavage. For example, M8 and M5 clearly differ greatly in their cleavage rates, yet both are single cleavers.

We selected the sgRNA that most efficiently targeted essential viral genes *in vitro*. Both m5 and m8 target essential genes for viral replication. Regarding meganucleases, cannot be generated as easily as screening various sgRNA, and thus meganucleases targeting additional sites are not currently available. We and Collectis (which created and provided the enzymes) have previously shown that both M8 and M5 have an impact on virus replication *in vitro* (Grosse et al 2011; Aubert et al 2016).

3. The comparison of a single meganuclease delivery to /CRSPR cas9 is still problematic; while they now compare cleavage using both ssAAV and the same serotype, the m5 is delivered with a TREX exonuclease while the CRSPR is not. What happens if a TREX is coexpressed with Cas9? For the most part TREX is not described in the text and should be. Also, the nature of the sgRNA guides might not be optimal, much as the meganucleases sometimes do not yield a high level of cleavage (m8 for example is a weak cleaver compared to M5). I do not understand why this point is still emphasized in the work. As they acknowledge there could be several variables that contribute, such as promoter, translation efficiency, sgRNA guide being optimal, and being able to access the presumably chromatinized DNA less efficiently than meganucleases.

Trex2 is a 3'-5' exonuclease, and is thought to promote increased gene editing of meganucleases by trimming the 3'-5' overhangs left after meganuclease cleavage. It is not expected to have any substantial effect when co-expressed with Cas9, since Cas9 cleavage generates blunt ends.

The following statement was added to the Results section:

Line 296-298: “The ssAAV construct carrying the HSV1m5 also deliver Trex2, a 3'-5' exonuclease that was shown previously to increase meganuclease gene editing²⁰.”

4. The abstract seems to contradict. Line 2 states the source of the recurrent disease is the peripheral nervous system, but line 21 the SCG is part of the autonomic system, no? this needs a correction.

The SCG is a component of the autonomic nervous system which is part of the PNS that controls involuntary body functions. Since we had to shorten the Abstract the statement mentioned by the reviewer has been deleted. However, we have modified the statement on line 46-48 in the Introduction section to indicate that both sensory and autonomic nervous systems are parts of the PNS: “After primary infection at the skin or mucosa, HSV establishes lifelong latency in both sensory (e.g. trigeminal and dorsal root ganglia) and autonomic (e.g. superior cervical and major pelvic ganglia) neurons of the peripheral nervous system.”

5. The use of scAAV and its likely importance in enhancing efficiency of transduction is vastly underplayed in this work.

The following statement was added to the Discussion to greater underline the higher transduction efficiency of scAAV compared to ssAAV:

Line 570-571: "It has been demonstrated previously that the use of scAAV vectors rather than ssAAV greatly enhanced the transduction efficiency³⁷."

6. *A rather annoying aspect is that the authors use of names (eg AAV CBh rather than denoting the serotype of the AAV as Rh10). This added to the difficulty in following the studies.*

Only in 1 place we had to use "AAV CBh" rather than denoting the serotype. This was done as multiple serotypes were used in the experiment described there, and it would have crowded greatly the figures to list each construct with the various serotypes used. Otherwise, the serotype(s) used in each experiment is/are indicated in the text (e.g. AAV8-CBh-m5) and corresponding figures as the reviewer would expect it. The only exception is for Supplementary Figure 1a, where different serotypes were used (AAV-8, PHP.S and RH10) but are indicated in the graphs in panels 1b-d;

We have modified Figure 8 where various combinations of AAV serotypes were used, and each serotype combination used in the experiment described there is now listed in the experimental schematic (Fig. 8a).

7. *Can the authors group the transduced neurons into a more descriptive terminology that would be familiar to the reader? Can they state if neurons groups transduced by specific AAV for the single cell analyses express certain markers (Trpv1, CGRP, NF200, etc.*

Our analysis did not allow grouping of the transduced neurons into a more descriptive terminology. We have stated this on Line 395-396: "Our analysis did not identify specific cellular transcripts unique to neurons transduced by specific AAV serotypes."

8. *Line 151 and 63 I think it would be pertinent to add a statement that SaCas9 is large and does not permit SCAAV to be developed.*

This point was made in the Discussion section when discussing the surprisingly low gene editing efficiency when using SaCas9.

Line 565-570: "The simplest explanation for this may be due to relative expression levels; due to its larger size Cas9 requires delivery by single-stranded (ss)AAV vectors, while meganucleases fit easily into the more transcriptionally efficient self-complementary (sc)AAVs, which do not require *de novo* second strand synthesis or intermolecular annealing for transgene expression."

However, we have now added the following statements:

Line 212-215: "Due to the large SaCas9 coding sequence (3.1 kb), ssAAV was used for the delivery system. The larger payload capacity of ssAAV allowed both SaCas9 and sgRNA expression cassettes to be on the same AAV construct, ensuring simultaneous delivery of SaCas9 and sgRNA to transduced cells."

9. *It is still not clear to me if reactivation is prevented or that virus emerging from reactivation is growth inhibited. The rebuttal explanation did not address this issue with clarity.*

This is a question we are very interested in addressing and will be important to answer. As we stated in our original rebuttal to the reviewers from *Nature*, we have performed several variations on initial experiments in an attempt to address this point. Unfortunately, due to the exceedingly low number of reactivating neurons following reactivation stress (on average 5-10 per ganglion) and variability between animals, we have not been able to unambiguously distinguish between the proposed mechanisms. Reaching a definitive answer will require a new set of studies, likely in a model system other than the mouse. Currently, we can say that meganuclease therapy leads to a reduction in *de novo* viral genome production, but we cannot distinguish whether it is due to a reduction in virus produced or a reduction in the number of the neurons harboring reactivating virus. This has been stated in the Discussion section line 525-527: "One limitation to

our study is that we cannot distinguish whether the observed reduction in *de novo* production of viral genomes resulted from a reduction in the number of reactivation events, or instead from a decrease in virus production after a reactivation event has initiated.”.

10. A point of discussion that they should address is the different routes of virus administration. The HSV goes through the eye corneal surface, yet the AAV are given via a different route. Otherwise the discussion was thoughtful, well done and nicely provoking.

We have added the following in the Discussion section:

Line 302-319: “The route of administration of the AAV delivery vectors influenced the efficiency of HSV gene editing. Delivery of AAV to the sites of HSV latency was shown previously to be less efficient after administration via the cornea (with or without scarification) than intradermal whisker pad (WP) injection²². In our optimization studies presented in Supplementary Fig. 1, we showed that while AAV delivery to ganglia was similar after injection via RO, TV and WP injection, RO led to the highest levels of gene editing, especially with Rh10 serotype.”